# Modified relaxation dynamics and coherent energy exchange in coupled vibration-cavity polaritons

A.D. Dunkelberger[1], B.T. Spann[1], K.P. Fears[2], B.S. Simpkins[2] & J.C. Owrutsky[2]

Coupling vibrational transitions to resonant optical modes creates vibrational polaritons shifted from the uncoupled molecular resonances and provides a convenient way to modify the energetics of molecular vibrations. This approach is a viable method to explore controlling chemical reactivity. In this work, we report pump–probe infrared spectroscopy of the cavity-coupled C–O stretching band of $W(CO)_6$ and the direct measurement of the lifetime of a vibration-cavity polariton. The upper polariton relaxes 10 times more quickly than the uncoupled vibrational mode. Tuning the polariton energy changes the polariton transient spectra and relaxation times. We also observe quantum beats, so-called vacuum Rabi oscillations, between the upper and lower vibration-cavity polaritons. In addition to establishing that coupling to an optical cavity modifies the energy-transfer dynamics of the coupled molecules, this work points out the possibility of systematic and predictive modification of the excited-state kinetics of vibration-cavity polariton systems.

[1] National Research Council Postdoctoral Associate, Chemistry Division, U. S. Naval Research Laboratory, Washington, District of Columbia 20375, USA. [2] Chemistry Division, Naval Research Laboratory, Washington, District Of Columbia 20375, USA. Correspondence and requests for materials should be addressed to J.C.O. (email: jeff.owrutsky@nrl.navy.mil).

**B**ond-specific molecular activation is a 'Holy Grail'[1,2] sought by physical chemists for imparting control over chemical reactions. Systematic manipulations of energy transfer and molecular reactions have been explored through laser-controlled chemistry, coherent control, ladder climbing and wavepacket sculpting[3–7]. Vibrational excitation can be an effective method for modifying reaction rates and product distributions in molecular beams and on surfaces, where individual vibrational eigenstates can be prepared and survive as species react[5,8–13]. However, efforts to achieve vibrational control in liquid- and solution-phase systems have achieved only limited success[14]. Coupling vibrational modes to optical cavities offers a method to systematically and predictably modify the vibrational energy landscape of a molecule and could finally lead to a generally applicable route to specific bond activation, which we term quantum optical chemistry.

A confined optical mode can couple to a resonant material transition and lead to enhanced absorption or emission rates, excited-state population control and, if the interaction energy exceeds the widths of the relevant modes, the formation of new hybrid states through normal-mode coupling[15–17]. The polariton states created through this coupling are separated in energy by an effective Rabi splitting, $\Omega = g(N^{1/2})$, where $g$ is the single-oscillator vacuum Rabi splitting and $N$ is the number of oscillators. Each hybridized state retains the quantum-mechanical character of both the optical cavity mode and the material excitation[15]. These hybrid states, known as cavity polaritons, have been used to create room temperature Bose–Einstein condensates, climb the Jaynes–Cummings ladder and tailor exciton transfer rates[18–20]. Though coupling between electronic transitions and cavity modes has been extensively studied[16,21–25], coupling between vibrational resonances and cavities has only recently been demonstrated. The independent works by Shalabney, et al.[26] and Long and Simpkins[27] report Fabry–Pérot (FP) cavity modes coupled to vibrational resonances in polymer films and were rapidly extended to vibrational polaritons in molecular liquids[28] and small molecules in solution[29]. Vibrational strong coupling has also recently been the subject of theoretical treatments by del Pino et al.[30,31] and recent calculations of two-dimensional infrared spectroscopy of vibration-cavity polaritons by Saurabh and Mukamel, the first prediction of time-domain spectroscopy on such a system[32]. Metamaterial surfaces have also been used to demonstrate coupling to vibrational modes, but the reduced quality factor associated with these meta-surfaces degrades the dispersive anti-crossings and makes it difficult to match the molecule and cavity linewidths[33,34]. Manipulating the energetics of a molecular vibration offers the potential to modify the molecule's reactivity, particularly if the coupled mode corresponds to a reactive coordinate on the molecule's potential energy surface. In fact, Hutchison et al.[22] have shown that modifying the energetics of electronic states through strong cavity coupling alters both the rate of photoisomerization and the work function[35] of spiropyran.

Transient spectroscopic techniques have revealed significantly modified dynamics in cavity-coupled electronic states[36] and direct energy transfer between the two cavity-coupled polariton states[24,37,38]. In the present communication, we report the transient pump–probe spectroscopy of a cavity-coupled vibrational mode. We find that cavity coupling greatly alters the transient response of vibrationally excited states. We couple the C–O stretching band of $W(CO)_6$ in hexane solution to a FP cavity mode and show (i) transient spectra dominated by excited-state transitions, (ii) excited-state relaxation rates that are an order of magnitude faster than in the uncoupled system and (iii) evidence for coherent energy exchange (vacuum Rabi oscillations) between hybridized vibration-cavity polaritons.

## Results

**Steady-state infrared transmission.** Vibration-cavity polaritons may be formed by placing a strong vibrational absorber between two closely spaced mirrors that form a FP cavity (shown schematically in Fig. 1a). For this work, the cavity resonance is angle-tuned through the triply degenerate antisymmetric C–O stretching band ($\alpha \sim 70{,}000\,\mathrm{M}^{-1}\mathrm{cm}^{-1}$ at $1{,}983\,\mathrm{cm}^{-1}$) of $W(CO)_6$ in hexane. The full dispersion of the molecule-loaded cavity (Fig. 1b) displays cavity modes that disperse with angle and anti-crossings as the cavity modes tune through resonance with the C–O stretching band (horizontal pink line). Multiple cavity modes are visible due to the cavity's long pathlength ($L = 25\,\mu\mathrm{m}$) and relatively narrow free spectral range ($150\,\mathrm{cm}^{-1}$). Longer cavities and higher-order modes yield the same coupling as first-order cavities ($L = \lambda/2n$, $n = 1$) if the absorber concentration is the same[28,29]. The normal-mode splitting is the minimum separation between the polariton modes and depends on the absorber concentration[15,29]. Transient measurements, discussed below, are performed at the angle of strongest coupling except where noted otherwise. Figure 1c shows the transmission spectrum of the sample at the angle of strongest coupling for a 10 mM solution of $W(CO)_6$ within a FP cavity. The two transmission features correspond to the upper- (UP, $1{,}994\,\mathrm{cm}^{-1}$) and lower-polariton (LP, $1{,}970\,\mathrm{cm}^{-1}$) branches, and are separated by the effective splitting, $\Omega = 24\,\mathrm{cm}^{-1}$ in this case.

**Vibrational relaxation of $W(CO)_6$.** To demonstrate how cavity coupling influences the energy-transfer dynamics of the C–O

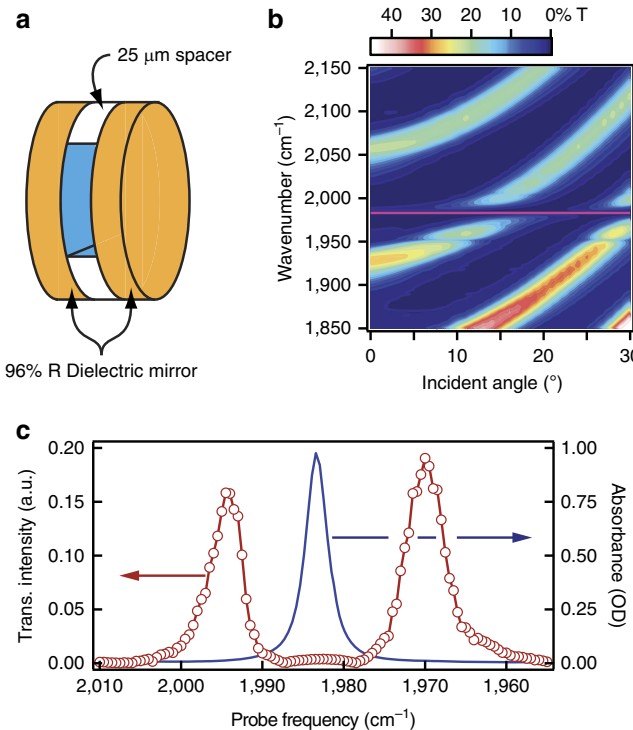

**Figure 1 | Experimental schematic and frequency-domain spectra.** (**a**) Schematic of sample comprising two dielectric mirrors separated by a PTFE spacer and filled with the solution of interest. (**b**) Dispersion of the infrared transmission of a cavity filled with 10 mM $W(CO)_6$ in hexane. The pink line corresponds to the position of the infrared-active C–O stretch at $1{,}983\,\mathrm{cm}^{-1}$. (**c**) Red points are the measured intensity of the probe pulse transmitted through the cavity in **b**, showing the UP and LP bands separated by the effective splitting. Blue trace is the Fourier transform infrared spectrum of $W(CO)_6$ without cavity coupling (5 mM solution in a $25\,\mu\mathrm{m}$ cell comprising $CaF_2$ windows), dominated by the C–O stretch absorption at $1{,}983\,\mathrm{cm}^{-1}$.

stretch, we use infrared pump–probe spectroscopy, with results for the uncoupled solution serving as a basis for comparison. We excite the systems with a short ($\sim$120 fs) infrared pulse whose spectrum is centred at the C–O stretch fundamental frequency (1,983 cm$^{-1}$, $\omega_{01}$) and is broad enough (100 cm$^{-1}$) to span both polariton modes in the coupled system. The subsequent changes in absorption are monitored with a second, spectrally resolved probe pulse whose delay with respect to the excitation pulse is varied. The transient infrared absorption of uncoupled W(CO)$_6$, displayed in Fig. 2, is a well-studied[39–42] and important guide to the interpretation of cavity-coupled results. We measure the transient absorption of uncoupled W(CO)$_6$ in a transmission geometry and report the signal in terms of $\Delta A = -\log_{10}(T/T_0)$. Immediately after excitation, the uncoupled system exhibits a negative transient absorption feature at the fundamental C–O stretching band frequency due to reduced population in the vibrational ground state, $v=0$, and increased population in $v=1$. We observe positive transient absorption features for the anharmonically shifted $v=2\leftarrow1$ ($\omega_{12}=1,968$ cm$^{-1}$) and $v=3\leftarrow2$ ($\omega_{23}=1,953$ cm$^{-1}$) hot-band transitions, shown schematically in the inset to Fig. 2b. Population of the $v=2$ state occurs because the anharmonic shift of the C–O stretch is relatively small (15 cm$^{-1}$) compared with the bandwidth of the excitation pulse ($\sim$100 cm$^{-1}$), and the W(CO)$_6$ interacts

with the excitation pulse more than once[42]. As the vibrational energy relaxes through intra- and intermolecular channels, the magnitudes of the transient absorption signals decrease. Time-resolved traces, shown in Fig. 2b and fit to a sequential kinetics model, reveal that the $v=2$ and $v=1$ populations decay with lifetimes of $\sim$70 and $\sim$140 ps, respectively, in good agreement with those reported in the seminal studies of W(CO)$_6$ relaxation[39,40].

**Transient spectroscopy of cavity-coupled W(CO)$_6$.** The transient spectra for the cavity-coupled system, presented for 3 and 100 ps after excitation in Fig. 3a, are markedly different from those of the uncoupled molecules. The data presented are reported in terms of $-\log_{10}(T/T_0)$ to connect to the majority of transient absorption studies conducted outside cavities. No ground-state bleach appears because the cavity does not transmit the probe pulse at 1,983 cm$^{-1}$. At early and late times, a large positive feature centred near the $\omega_{12}$ transition frequency, 1,968 cm$^{-1}$, dominates the spectra. This region coincides with the LP branch. On the low-energy edge of this region, below 1,961 cm$^{-1}$, the response evolves from positive to negative in tens of picoseconds (compare red with blue data in Fig. 3a). This negative signal is unique to the cavity-coupled measurement. The smaller responses in the UP region near 2,005 cm$^{-1}$, which are completely absent in the uncoupled case, decay monotonically.

In the uncoupled system, new absorptive features are readily explained as excited-state transitions and their absorption intensities are linear with excited-state population. Inside the cavity, however, modified populations can lead to qualitative changes in the spectral shape (for example, reduced splitting or additional splittings at new frequencies). Analysing the cavity-coupled results requires a more sophisticated approach than outside the cavity. Briefly, we use an analytical expression for FP cavity transmission, $T_{cav}$, which treats the oscillators as an effective medium to calculate transient spectra with excited-state population[15,27,43].

$$T_{cav} = \frac{T^2 e^{-\alpha L}}{1 + R^2 e^{-2\alpha L} - 2Re^{-\alpha L}\cos(4\pi n L\bar{v} + 2\varphi)} \quad (1)$$

The parameters $R$ and $T$ are the reflectance and transmittance, respectively, of each mirror (not to be confused with the amplitude reflectance and transmission coefficients, often designated $t$ and $r$), assumed to be constant over the 200 cm$^{-1}$ region of interest. $L$ is the cavity length, $\varphi$ is a phase shift that occurs as light reflects from each mirror, $n$ and $\alpha$ are the frequency-dependent optical index and absorption coefficient (that is, absorption per unit length, $\alpha = 2\pi k v$, where $k$ is the imaginary part of the complex refractive index), respectively, of the intracavity medium. We include the presence of ground- and excited-state oscillators by treating them as Lorentz oscillators in $n$ and $\alpha$ (detailed in Supplementary Note 1 and Supplementary Fig. 1). In cavity-coupled systems $\Delta A$ is no longer directly connected to differential transmission and, instead, requires the measure of both transmission and reflection[37]. However, we find that the transients extracted from transmission and absorption are in reasonable agreement and include these results in Supplementary Note 2, Supplementary Figs 2 and 3 and Supplementary Table 1.

The choice of oscillators and their frequencies is critical for reproducing the experimental results. As detailed below, we find that equation (1) adequately describes the data when we assign population to a quickly decaying upper polariton and comparatively long-lived reservoir $v=1$ and LP states. These assignments are further supported by changing the polariton energies through angle tuning (see below). Figure 3b schematically shows the energy levels of the cavity-coupled system before and after the pump

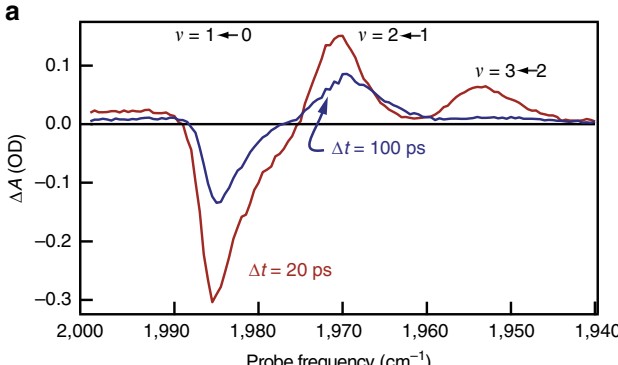

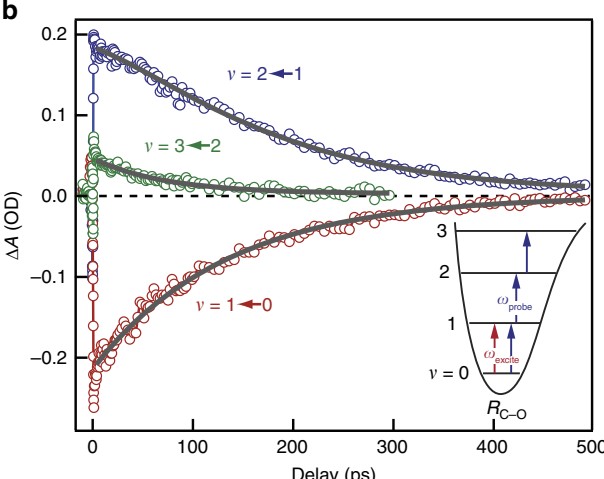

**Figure 2 | Transient spectroscopy and kinetics of uncoupled W(CO)$_6$.** (**a**) Transient absorption spectra of uncoupled W(CO)$_6$ in hexane measured 20 (red) and 100 ps (blue) after infrared excitation. (**b**) Kinetic traces measured at the peak of the ground-state bleach (1,983 cm$^{-1}$) and first (1,968 cm$^{-1}$) and second (1,953 cm$^{-1}$) hot-band transitions. Solid grey lines are results of a fit to the relaxation dynamics. The inset schematically shows the transitions excited and probed in the experiment.

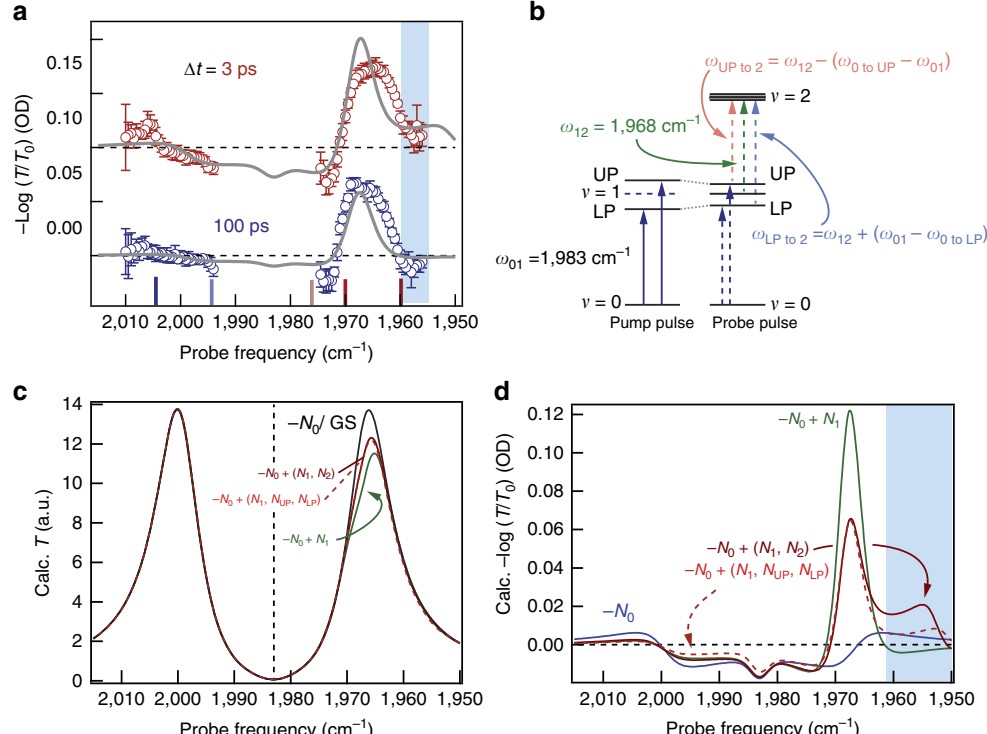

**Figure 3 | Transient spectroscopy of cavity-coupled W(CO)₆.** (**a**) Transient spectra of cavity-coupled $W(CO)_6$ (20 mM in hexane) measured 3 (red) and 100 ps (blue) after excitation. Open circles are measured $-\log_{10}(T/T_0)$ and error bars represent one s.d. obtained from 10 measurements at a given probe frequency. Solid grey lines are models detailed in the text. Traces are offset for clarity. Coloured vertical lines on the frequency axis correspond to probe frequencies presented in Fig. 4. The blue box highlights the low-frequency response that indicates UP population. (**b**) Schematic representation of energy levels and transitions involved in the cavity-coupled transient absorption experiment. The pump pulse excites the UP and LP levels. The probe pulse monitors the reduced splitting between UP and LP, as well as additional transitions from UP, LP and $v=1$ to 2. The excited-state population is so low that the splitting in $v=2$ is smaller than the absorption linewidth. (**c**) Calculated transmission spectra for varying amplitude (population) of oscillators at the frequencies described in **b**. The ground-state (black) curve has population only in $v=0$ and is calculated for 20 mM $W(CO)_6$. Other curves are calculated for a 1% reduction in population at $v=0$ (labelled $-N_0$) and commensurately increased population in other modes (for example, $+N_1$ for increased population in $v=1$). The calculated spectrum for a 1% depletion in ground-state population, $-N_0$, is indistinguishable from the GS curve. (**d**) Calculated differential absorption spectra for the same population distributions used to calculate the spectra in **c**. Population in either $v=2$ or UP and LP gives rise to qualitatively similar spectra. The blue box highlights the low-frequency response that indicates UP population.

pulse. The excitation pulse promotes ground-state $W(CO)_6$ to the LP and UP. This reduction in ground-state population leads to reduced absorption and, therefore, reduced splitting, similar to phenomena observed in exciton polaritons[37,38,44,45]. The probe pulse measures the excited-state spectrum of the non-equilibrium distribution of ground state, LP-, UP- and reservoir-excited $W(CO)_6$. We use the term reservoir to describe vibrational modes that are not coupled to the cavity, adopting the language and description from the exciton-coupling literature[19,46,47]. As discussed below, predictions from Eq. 1 match the experimental transient spectra when we use excited-sate absorption frequencies corresponding to transitions from the excited states to $v=2$. The magnitude of the transient response indicates that the excitation pulse excites only ∼1% of molecules. Because the splitting depends on $N^{1/2}$, absorption from the excited polariton or reservoir states to $v=2$ results in only a minimal splitting of $v=2$ (∼2–3 cm$^{-1}$).

Calculations of the transient spectra, obtained by varying populations in ground and excited states in equation (1), provide good agreement with the observed spectra. First, the transmission spectrum for the ground-state system (grey curve in Fig. 3c) is modelled by including only a single oscillator at $\omega_{01}$ with an amplitude, $A_{01}$, chosen to yield the experimentally determined splitting ($\Omega = 32$ cm$^{-1}$ for a 20 mM $W(CO)_6$ solution). Next, excited-state spectra are calculated by incorporating transitions

from UP, LP or reservoir $v=1$ states to the $v=2$ state (that is, by reducing $A_{01}$ and increasing $A_{12}$, $A_{LP\,to\,2}$ and/or $A_{UP\,to\,2}$, details in Supplementary Note 3). The total oscillator amplitude in the system is conserved and we assume that the polariton excited-state absorption transition intensity is the same as for the reservoir transition. The differences between ground- and excited-state transmission spectra, converted to $-\log_{10}(T/T_0)$, yield the predicted transient spectra (Fig. 3d). With only ground-state depletion (no absorption from excited states, blue curve) the spectrum is symmetric about the $\omega_{01}$ vibrational resonance and reflects a slight reduction in the effective splitting due to reduced population in the ground state[29,44]. The calculated spectrum for a ∼1% depletion in ground-state concentration matches the experimental spectrum in the UP region but fails to capture the behaviour in the LP region.

Reservoir $v=1$ population has to be included (green curve in Fig. 3d and green arrow in Fig. 3b) to reproduce the dominant positive feature that appears at the $\omega_{12}$ transition frequency, along with the negative response at the low-energy edge of the LP branch, below 1,961 cm$^{-1}$. This negative feature is a unique to the cavity-coupled transient response and reports on the weak coupling between $\omega_{12}$ and the cavity. Again, because $\Omega$ scales with the square root of absorber concentration[15,27–29], we expect the splitting of $v=2$ due to coupling of the excited-state transitions to be 2–3 cm$^{-1}$. The splitting due to the weak

coupling, depicted in Fig. 3b, is smaller than the molecular or cavity widths, but nonetheless is observable and induces increased transmission below $1{,}961\,cm^{-1}$. This calculated spectrum reproduces the experimental spectrum measured 100 ps after excitation.

Capturing the early-time transient spectrum, particularly the positive feature below $1{,}961\,cm^{-1}$, requires the inclusion of additional transitions. Choosing the additional transitions is somewhat ambiguous because half the effective splitting, $\Omega/2$, is close to the anharmonic shift of $W(CO)_6$, making $\omega_{UP\ to\ 2}$ close to $\omega_{23}$ (reservoir $v=2$ to 3, $1{,}953\,cm^{-1}$) and $\omega_{LP\ to\ 2}$ close to $\omega_{01}$. Specifically, for a 20 mM solution at the angle of strongest coupling, $\omega_{UP\ to\ 2}=1{,}951\,cm^{-1}$ and $\omega_{LP\ to\ 2}=1{,}985\,cm^{-1}$. We consider two cases in Fig. 3d, one with reservoir $v=2$ populated (solid red) and one with the UP and LP populated (dashed red). Both calculated spectra show a positive response at the low-energy edge and are qualitatively consistent with the experimental spectrum at 3 ps (compare red data in Fig. 3a with red curves in d). So little light is transmitted through the cavity at these lower frequencies that we cannot reliably determine the centre frequency of this low-energy absorptive feature, but we tentatively assign it to $\omega_{UP\ to\ 2}$ because its decay rate and centre frequency depend on the polariton tuning (see angle-dependent results below). Because the transition at $\omega_{LP\ to\ 2}$ nearly overlaps $\omega_{01}$, it has only subtle effects on the transient spectra (Supplementary Note 4 and Supplementary Fig. 4).

In summary, the excellent agreement between the calculated and experimental spectra shows that the optically excited cavity-coupled system can be adequately described by a classical treatment of a FP cavity containing a distribution of excited-state absorbers. The simplest model that achieves agreement includes two excited-state transitions: UP to 2 and reservoir $v=1$ to 2. We reason that the LP should also be populated, but the LP has no distinct spectral signature. Our treatment shows that UP population decays quickly, leaving reservoir $v=1$ and LP population that decays more slowly, and successfully reproduces the experimental results that are markedly different from those measured outside a FP cavity.

**Time-dependence of cavity-coupled transient absorption.** By measuring transient transmission decays at specific probe frequencies, we obtain quantitative relaxation rates that can be compared with the uncoupled case. In the first few picoseconds after excitation, the response at all probe frequencies is dominated by large oscillations. We assign these early-time dynamics to vacuum Rabi oscillations—coherent energy transfer between the polariton branches—and discuss them further below. After these oscillations decay, we examine the time-dependence of the transmission, shown in Fig. 4, at probe frequencies that span the UP and LP branches (coloured ticks in Fig. 3a correspond to frequency positions of similarly coloured data in Fig. 4). Depending on the probe frequency, the decay is either monotonic or has two decay components. Because the shape of the transient spectra changes with evolving excited-state population, each probe frequency reports on multiple species. We assign the fast component to UP decay based on the previous discussion of the spectral evolution and the angle-dependence of its lifetime. We infer that the slow decay process involves reservoir $v=1$ decay and LP decay (Supplementary Note 4).

We fit the time evolution of the transient response to a biexponential decay, using a global fitting technique that requires the two time constants to be the same for all the decay curves. Global fitting (parameters included in Supplementary Table 2) yields population lifetimes: $\tau_{UP}=k_{UP}^{-1}=22\,ps$ and $\tau_{slow}=k_{slow}^{-1}=170\,ps$. The value determined for $k_{UP}$ represents

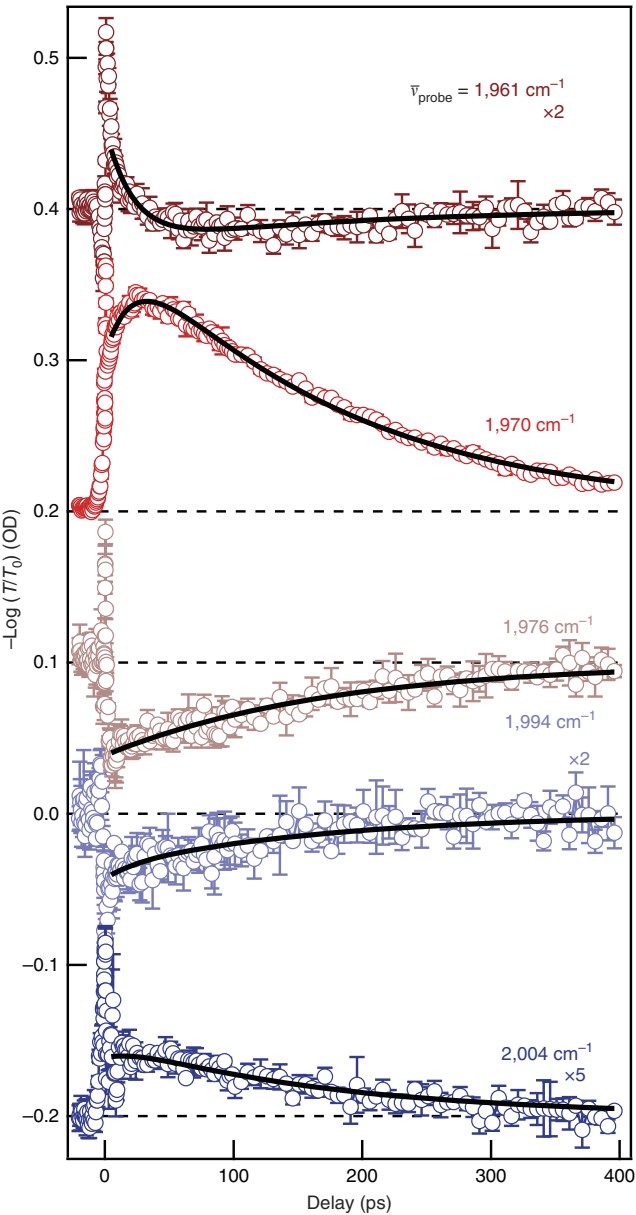

**Figure 4 | Relaxation kinetics of cavity-coupled W(CO)₆.** Transient response of cavity-coupled $W(CO)_6$ as a function of delay between excitation and probe pulses. Open circles are the average of three traces at each indicated probe frequency. The traces are offset and, where indicated, scaled for clarity. Solid lines are global fits to a biexponential kinetics model that yields lifetimes of 22 ps for the fast component (upper polariton) and 170 ps for the slow component (reservoir $v=1$ and LP). Error bars represent s.d. from three consecutive scans.

the first direct measurement of a vibrational polariton lifetime to the best of our knowledge, as substantiated by the angle-dependent results presented below, and is nearly an order of magnitude faster than the uncoupled $v=1$ relaxation. We surmise that the optical-mode character of the polariton speeds the vibrational energy relaxation, as discussed in more detail below. The slow decay is non-exponential, but its 1/e lifetime is increased 20% compared with $v=1$ decay in uncoupled $W(CO)_6$. We take this as evidence that slowly decaying LP population contributes to the signals we observe. We estimate 15% uncertainties in these values based on reproducibility in repeated measurements.

We estimate the excited-state populations by modelling the transient spectra measured 3 and 100 ps after excitation (Fig. 3a) using equation (1). Using the time constants obtained from the global fits, we calculate transient spectra for different initial concentrations of excited-state absorbers. We find agreement with the experimental signal magnitudes, especially those of the features at 1,968 and 1,961 $cm^{-1}$, when we assume 1% total excitation with $N_{UP}$:$N_{\nu=1} = 1{:}2$. The parameters for one such calculation (grey solid lines, Fig. 3a) are included in Supplementary Note 5. The agreement in the signal magnitude suggests initial excitation on the order of 1%, which is quite low compared with measurements of uncoupled $W(CO)_6$, where up to 30% of the molecules are promoted to the $\nu=1$ state at similar excitation intensities. The low level of excited-state population is probably due to radiative losses during the early-time vacuum Rabi oscillations, as discussed below, but could also be associated with how the cavity-coupled transmission affects how the system is excited. Nevertheless, the transient spectra show large signals even for low excited-state populations due to cavity-enhanced absorption at the resonant transmission peaks. In the uncoupled system, exciting 1% of the $W(CO)_6$ to $\nu=1$ would result in $-\log_{10}(T/T_0) = 0.01$, an order of magnitude smaller than we observe in the cavity. If coupling to the cavity modifies the excited-state absorption intensity of the polaritons compared with the reservoir, our estimate of the polariton population will be subsequently modified.

**Angle-dependent polariton tuning**. As mentioned above, one potential complication in our interpretation arises because $\omega_{UP\,to\,2}$ is close to the reservoir $\nu=2$ to 3 transition frequency, $\omega_{23}$, making its assignment potentially ambiguous. We use the angle dependence of polariton energy to distinguish between polariton and reservoir populations and to substantiate our assignment of the quickly relaxing absorber to UP-excited $W(CO)_6$. Changing the cavity angle systematically tunes the polariton energies while not affecting the reservoir vibrational state energies. Consequently, changing the angle will modify the position of $\omega_{UP\,to\,2}$ but not $\omega_{23}$. Figure 5a shows transient spectra for cavity-coupled 10 mM $W(CO)_6$ at two different incidence angles. At 17° (blue), the transient spectra resemble those previously presented (Fig. 3a). At 13°, that is, red detuned from strongest coupling (red curve), the spectra show marked, qualitative differences. At this detuning, $\omega_{UP\,to\,2}$ has been shifted ~7 to 1,961 $cm^{-1}$ but the $\omega_{23}$ transition does not depend on the cavity detuning and remains at 1,953 $cm^{-1}$. Under these conditions, we observe a strong negative transient response at 1,953 $cm^{-1}$ 5 ps after excitation (open circles), which is all but absent after 100 ps (filled circles). If the fast component resulted from reservoir $\nu=2$ population, we would observe a positive response at 1,953 $cm^{-1}$, as opposed to the negative feature we observe, and so we assign the fast component to population in UP. Further evidence for this assignment is shown in Fig. 5b, where calculated transient spectra are shown for population in either the UP or reservoir $\nu=2$ levels overlaid on the experimental early-time spectrum taken under red detuned conditions. The calculated spectrum that includes UP population (dotted green) captures the positive features at 1,961 and 1,966 $cm^{-1}$ and the negative feature at 1,954 $cm^{-1}$ though the feature amplitudes and the positive feature at the lowest probe energy ($<1,950\,cm^{-1}$) are not accurately predicted. Calculating the spectrum with population in uncoupled $\nu=2$ (solid green) results in a positive response at ~1,953 $cm^{-1}$ and does not even qualitatively agree with the data. The calculated spectrum for only reservoir $\nu=1$ population (blue) agrees well with the experimental spectrum at $\Delta t = 100$ ps (filled circles). Again, the $\omega_{LP\,to\,2}$ (1,983 $cm^{-1}$ for 17° and 1,992 $cm^{-1}$ for 13°) transition

is close to $\omega_{01}$ (1,983 $cm^{-1}$) and so has only subtle effects on the calculated and observed spectra (as described in Supplementary Note 4).

The direct measurement of angle-dependent lifetimes further supports our interpretation. Figure 5c shows transient absorption at 1,968 $cm^{-1}$ measured at two different incidence angles and lifetimes extracted from global fit results (tabulated in Supplementary Table 3) for each angle. At 29° (strongest coupling; blue trace; UP = ,1994 $cm^{-1}$, LP = 1,967 $cm^{-1}$), the UP lifetime is 21 ps, but at 27.5° (red detuned; red trace; UP = 1,991 $cm^{-1}$, LP = 1,961 $cm^{-1}$), the lifetime increases to 50 ps. We do not expect the lifetime of uncoupled reservoir population to depend on the incidence angle. However, the UP lifetime could vary with incidence angle for two reasons. First, the polaritons have varying amounts of vibration and cavity character as they are tuned[48,49]. The UP becomes more vibration-like as the cavity is red detuned and consequently should decay more slowly since the uncoupled vibration is longer-lived than the cavity photon. This is consistent with the lifetimes we observe. Second, the polaritons are shifting in energy relative to nearby acceptor states. For high-frequency modes, vibrational relaxation generally proceeds through lower-frequency intra- and intermolecular gateway modes before dissipating to low-frequency collective motion of the environment[50–52]. The process is so sensitive to the density of states of the solvent environment and other ligands bound to the metal that the population relaxation lifetime can vary by an order of magnitude with different solvents and co-ligands[50,53,54]. In metal carbonyls, these acceptor modes can include slightly higher-lying Raman active modes, low-frequency bending modes of the metal carbonyl, localized solvent vibrations and collective solvent motion[40,50–54]. Tuning the energy of the polariton might bring the polariton closer to or farther from resonance with an acceptor mode, thus changing its lifetime. Regardless of mechanism (altered polariton character or energy), the polariton mode formed by coupling the first excited state of the asymmetric C–O stretch has a lifetime 10 times shorter than that of the uncoupled C–O stretch. The longer decay rate component appears to decrease from 200 ± 30 to 160 ± 20 ps with red tuning, further suggesting there is a contribution from LP population to the slow-decaying spectrum. These results represent the first measurements of the population lifetime of a vibration-cavity polariton, to the best of our knowledge, and are an important benchmark for other experiments that might utilize excited polaritons.

**Oscillations in early-time dynamics**. In systems that are strongly coupled, including cavity-coupled electronic excitations in semiconductors and plasmon polaritons coupled to organic dyes, early-time oscillations, whose frequency scales directly with effective splitting, have been observed and attributed to vacuum Rabi oscillations, that is, coherent exchange of energy between the coupled states[15,45,55]. Figure 6a shows examples of the oscillatory signals we observe. The oscillations start at negative time delays when the probe pulse arrives at the sample before the excitation pulse, suggesting that the probe pulse generates a coherence that the excitation pulse interacts with, as observed in perturbed free-induction decays[45,56–58]. However, the periods of the oscillations we observe are insensitive to probe frequency or pump pulse intensity, which is in stark contrast to the highly probe-frequency dependent perturbed free-induction decays or uncoupled Rabi oscillations ('flopping')[59]. Importantly, the frequency of the oscillations observed in the vibration-cavity polariton system scales linearly with, and is nearly equal to, the measured splitting (Fig. 6b,c), which we control by changing the $W(CO)_6$ solution concentration. This equivalency is an

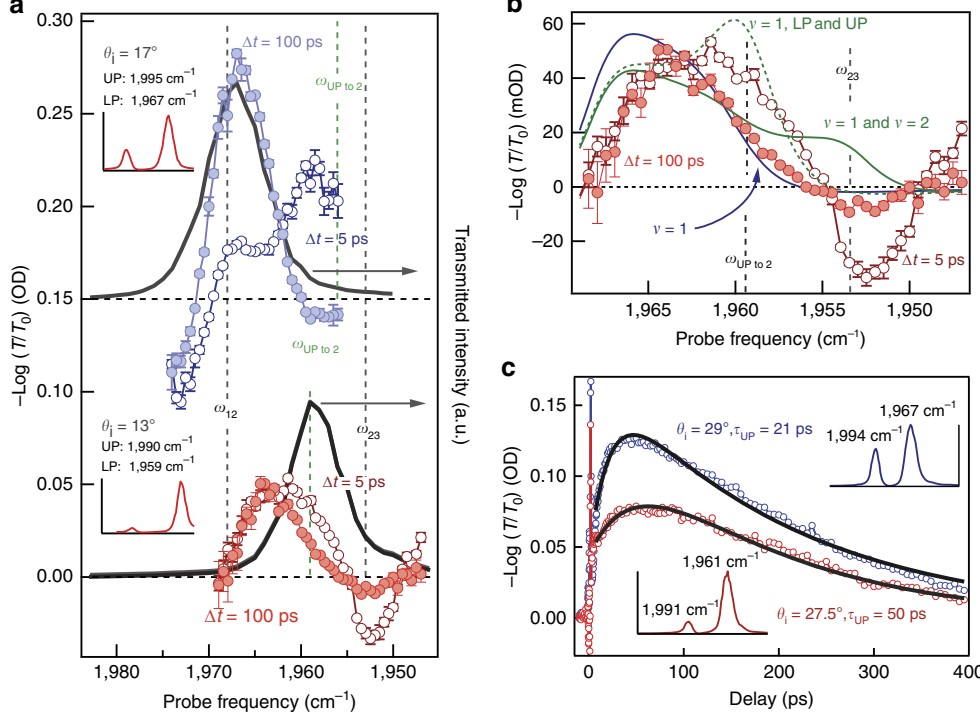

**Figure 5 | Angle dependence of transient spectroscopy and kinetics.** (**a**) Transient spectra measured 5 (open circles) and 100 ps (filled circles) after excitation at 17° and 13° for a sample comprising 10 mM W(CO)$_6$ in a FP cavity. Solid grey lines are transmitted probe light measured at the detector and show the steady-state transmission of the cavity at a particular angle of incidence. Dashed lines mark the frequencies of excited-state transitions involving reservoir (black) and polariton (green) levels. Error bars represent s.d. from five measurements at each probe frequency. (**b**) Expanded view of the transient spectra measured at 13°. Solid blue line is a calculated transient spectrum for population in reservoir $v = 1$, analogous to the calculation in Fig. 3. Solid green line is the calculated spectrum with population in reservoir $v = 1$ and 2, while dashed green line is the calculated transient spectrum for population in $v = 1$ and the polariton levels. Error bars represent s.d. from five measurements at each probe frequency. (**c**) Angle-dependent decay kinetics measured at 1,968 cm$^{-1}$. The same sample was interrogated at 27.5° (red) and 29° (blue). Insets are the transmission spectrum measured at each angle with maxima labelled.

important criterion for identifying vacuum Rabi oscillations and has been used to attribute similar oscillatory behaviour in cavity-coupled excitonic systems to vacuum Rabi oscillations[24,45]. This notable observation is the first evidence for vacuum Rabi oscillations in a coupled vibration-cavity polariton system, to the best of our knowledge. We note that interference between two fields at the sample can also create such oscillations[15], but measurements where the probe pulse is either parallel or perpendicular to the excitation pulse give identical oscillations, making us confident that the oscillations are indeed Rabi oscillations. Higher-order nonlinear spectroscopies can differentiate between mechanisms of coherent energy transfer and present an exciting avenue for future experiments[60].

These vacuum Rabi oscillations represent coherent energy transfer between the UP and LP and, as such, exhibit a dephasing rate that is the average of the dephasing rates of the coupled modes. For the system studied here, we estimate dephasing rates for the cavity modes and molecular vibrations from their homogeneous linewidths and find ∼3 (10 cm$^{-1}$) and 9 ps (3 cm$^{-1}$), respectively. We suggest the relatively high loss associated with the cavity photons diminishes the vibrationally excited population during the oscillation dephasing time. This phenomenon has been recognized in other cavity-coupled systems[61] and may account for the surprisingly low level of excited-state population (∼1%) observed after these oscillations decay. We note the oscillation period for vibration-cavity polaritons (∼ps) is substantially longer than for exciton polaritons (∼10's fs)[45]. This is a consequence of the much smaller splitting involved in vibrational modes and may facilitate

experiments on processes that occur during the oscillations and before they dephase.

## Discussion

We observe the signature of excited-state absorption from a vibration-cavity polariton mode, the first such report, to the best of our knowledge, and find that the polariton relaxes an order of magnitude more quickly than the uncoupled C–O stretch. Angle-tuning the cavity allows for systematic modification of the polariton lifetime either due to modification of polariton character (vibration versus cavity photon) or energy proximity to acceptor modes. Potential evidence of prompt population loss as a result of cavity leakage was observed and may offer a means to enhance or reduce excited-state population in a systematic way. The transient spectra of cavity-coupled W(CO)$_6$ are qualitatively different from uncoupled W(CO)$_6$ in that the sensitivity to excited-state population is greatly enhanced and new spectral features appear in spectral regions defined by the polariton branches. A classical description of the cavity transmission captures these differences and supports our assignment of the new features as signatures of excited-state absorption from reservoir and polariton modes. At early times, we observe oscillations that we attribute to coherent energy transfer between the UP and LP modes and confirm that the frequency of these vacuum Rabi oscillations corresponds to the measured effective splitting. This notable result is evidence of an ensemble of vibrational polaritons responding coherently after infrared excitation.

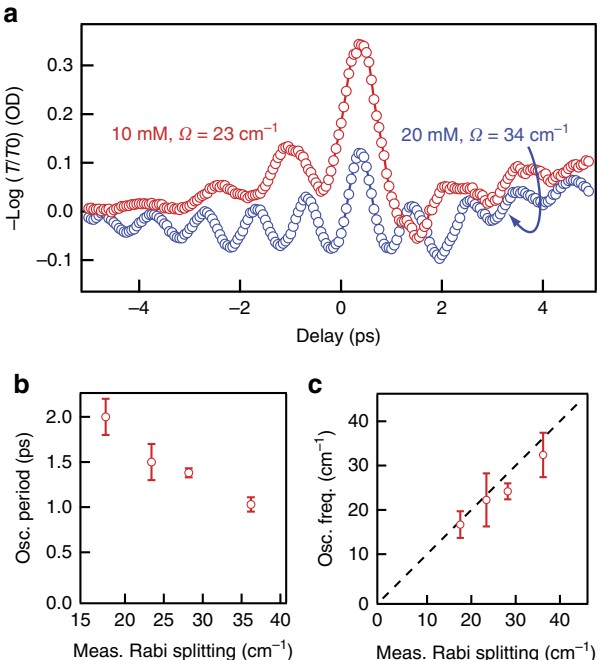

**Figure 6 | Vacuum Rabi oscillations in cavity-coupled W(CO)₆.**
(**a**) Representative transient absorption traces highlighting the strong oscillatory signals evident when the pulses are nearly coincident in time. (**b**) Measured period of early-time oscillations as a function of the effective splitting extracted from the dispersion curve for each concentration. (**c**) Frequency calculated from the oscillation period for cavity-coupled W(CO)₆ of varying concentration as a function of the measured splitting. The dashed line represents the case where the oscillation frequency and effective splitting are equal. Error bars represent s.d. from measurements of the period across each kinetic scan.

Since this is, as far as we know, the first experimental report of time-resolved infrared spectroscopy of a cavity-coupled vibrational system, we highlight some salient features. First, vibrations are anharmonic, causing the excited-state transitions to be red-shifted relative to the ground-state transition. When the anharmonic shift corresponds to half the effective splitting, the hot-band absorption coincides with the LP branch and dominates the transient signal. On the other hand, excitons do not typically exhibit anharmonically shifted excited states. Transient absorption for cavity-coupled excitons typically exhibits reduced effective splitting ($\Omega$) due to ground-state bleaching.

The simple biexponential kinetic model we use is somewhat surprising in its effectiveness. In other polariton systems, incoherent energy transfer between the polariton modes and reservoir modes has been observed[19]. In our case, the congested spectra preclude us from distinguishing between these pathways. Each excited-state absorption transition couples to the cavity, leading to overlapping spectral responses. There is no spectral region where the transient response reports only, for instance, on the reservoir population, so we cannot distinguish between the model presented above (UP and reservoir both decay independently) or a decay path in which the UP decays to the reservoir. Other molecular systems where the anharmonicity is very different from experimentally accessible splittings might give clear evidence of more complicated dynamics.

Cavity-coupled vibrational systems also show several key differences from uncoupled vibrational measurements. Several groups have shown that modifying the spectrum of the excitation pulse can change the population of higher excited states[40,42,62]. The polaritons produced by cavity coupling dictate the spectrum

of the excitation pulse in this experiment, in that only light that the cavity transmits can interact with the absorbers contained within. This spectral filtering could lead to unforeseen enhancement of up-pumping in the cavity, but we observe no direct evidence of such enhancement.

Strong coupling between optical modes and vibrations may provide a way to control chemistry by selectively and systematically modifying population relaxation lifetimes and, therefore, non-equilibrium vibrational population distributions[26]. Thomas *et al.*[64] have recently demonstrated modification of the kinetics of a chemical reaction when reactant and product vibrational modes are coupled to a cavity. Another way to achieve vibrational control could be to prepare vibrationally excited reactants and then initiate a photoreaction with second laser pulse. The time delay between the preparation and initiation pulses would define whether reactants were in a coherent vibration-cavity polariton state or a purely vibrational state during reaction thus allowing one to probe reactivity of the hybrid polariton. We further suspect that initiating the chemical reaction before the polariton decays could have a much different effect on a system's reactivity than relying on modification of the relaxation of purely vibrational modes.

## Methods

**Cavity construction.** The cavity comprises two dielectric mirrors with ∼96% reflectivity separated by a nominally 25 μm polytetrafluoroethylene (PTFE) spacer and filled with W(CO)₆ solutions of varying concentration between 5 and 20 mM. The mirrors are 25 mm × 2 mm CaF₂ flats coated by Universal Thin Film Labs. We specified the proprietary coating to give 92 ± 2% reflectivity between 4.7 and 5.1 μm at normal incidence, transmission in the visible (coating is yellow to the eye) and compatibility with water and organic solvents. The etalon formed by the empty cavity has a free spectral range of ∼140 cm⁻¹, a fringe width of ∼10 cm⁻¹ and a finesse of 14 (Supplementary Fig. 5 and Supplementary Note 6). The free spectral range indicates that the cavity length is about 30 μm. Assembled cavities can vary in length resulting in variations in the angle of strongest coupling from cavity to cavity. The measured cavity splitting varies with cavity angle and absorber concentration, leading to small variations from sample to sample. In all cases, we accompany transient results with the steady-state splitting we measure for that specific sample.

**Ultrafast laser instrument.** The ultrafast laser system has been described previously[63]. Briefly, an optical parametric amplifier converts 1 mJ, 120 fs pulses centred at 800 nm to 4 μJ pulses centred at 1,983 cm⁻¹. A CaF₂ window reflects 4% of the infrared pulse to a computer-controlled delay line to be used as a probe. The remainder of the infrared pulse is mechanically chopped and used to excite the sample. The excitation and probe pulses intersect the sample at the same small angle with respect to the sample normal but in the same vertical plane (perpendicular to the instrument table). We rotate the sample about the axis perpendicular to the table to ensure that both pulses intersect the sample at the same angle. For uncoupled measurements, the probe pulse was polarized at the 'magic angle' (54.7°) to the excitation pulse to avoid anisotropic contributions to the signal. In the cavity-coupled case, we observe no significant anisotropy. After the sample, a monochromator selects probe frequencies that we detect with a mercury–cadmium–telluride sensor. LabVIEW software controls the delay stage, monochromator and lock-in amplifiers used for data acquisition.

**Data availability.** The data that support the findings of this study are available on request from the corresponding author J.C.O.

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

## Acknowledgements

The authors thank Drs J.P. Long, H.D. Ladouceur, A. Baronavski, T. Reinecke, G. Beadie and O. Soykal for enlightening conversations. A.D.D. and B.T.S. gratefully acknowledge Research Associateships administered by the National Research Council. This work is supported by the Office of Naval Research through the Institute for Nanoscience at the US Naval Research laboratory.

## Author contributions

All authors contributed equally to manuscript preparation. A.D.D. carried out the transient measurements and analysis; B.T.S. assisted in transient measurement and analytical understanding; K.P.F. prepared and advised on chemical systems; B.S.S. developed and applied the analytical treatment and advised on interpretation; J.C.O. oversaw the project and advised on experimental design and interpretation.

## Additional information

**Competing financial interests:** The authors declare no competing financial interests.

**How to cite this article**: Dunkelberger, A. D. *et al.* Modified relaxation dynamics and coherent energy exchange in coupled vibration-cavity polaritons. *Nat. Commun.* **7**, 13504 doi: 10.1038/ncomms13504 (2016).

**Publisher's note**: 

