## [Peer Review File · Nature Communications]

Reviewers' comments:

Reviewer #1 (Remarks to the Author):

The authors present the first pump-probe study of vibro-polaritons which display impressive Rabi-oscillations at short time scales and overall modification of the dynamics. The study is interesting, sound, well written and definitely merits publication in Nature Comm., Nature Physics or PRL. However there are some points that need clarification. Below are some specific points and comments:

1) The most important point is the way the transient spectra are recorded which is not indicated in the text. It appears that the absorbance are only acquired from the delta Ts.

As was shown in ref 47, the transient Absorption spectra of sample inside cavities (whether strongly coupled or not) can only be measured accurately by recording both the reflection and the transmission. Relying on a fit using eq. 1 is not enough.

2) In small cavities, the large surface to volume ratios can perturb dynamics. The authors should do a control in a detuned cavity. I don't expect any change but since this is the first study of this kind, I think it is important that all the controls are done.

3) The possibility of weak coupling effects which could influence the results can not be excluded. It would be really interesting if the dynamics were studied at low enough concentration to avoid strong coupling but in resonance with the cavity.

4) The authors must provide information on the composition of the mirrors and cavity response (Q factor).

5) The notion that there is a large reservoir of uncoupled molecules is not shared by everyone in the community. There are always some uncoupled vibrations for simple reasons such as field strength and orientation versus the field. Hence I find the conclusion of the discussion on p12 very reasonable.

6) minor points:

- not everyone is familiar with mODs (p9 last line), so perhaps it is better just to stick to the word absorbance, e.g. 10^{-2} absorbance units.

- The direct excitation to $v=2$ is not formally allowed by selection rules (p4), perhaps a sentence explaining why it is possible to populate this level in your system would be welcome, e.g. sequential excitation?...

Reviewer #2 (Remarks to the Author):

This paper reports new experimental results in the promising field of vibrational polaritons. The authors performed infrared pump-probe experiments on samples exhibiting strong coupling between vibration and cavity modes. Key results are measurements of transient spectra at the femtosecond time scale and spectrally resolved kinetics, allowing an analysis of polariton dynamics. These original results highlight the effects of polaritons on molecular dynamics. This kind of novel experiments opens the way to novel chemistry with a possible control of chemical reactivity by coupling vibrational modes with cavity modes. In that sense, this work presents a great interest. The context is well introduced and the paper is clearly written.

The T1u mode of $W(CO)_6$ has a very large transition dipole moment and seems a good choice to get polaritons in the strong coupling regime. Vibrational dynamics in different solvents out of a

cavity was extensively studied, bringing a solid background to the present work. The quality of data as shown in the figures is very good.

However, the analysis is not convincing enough to be published in the present form. It impacts the reliability of the conclusions.

First of all, several important quantitative data (results and parameters in the fits) are missing - to be added either in the manuscript or in the Supplementary Information. I detail them below. The kinetic model is especially not convincing. The assumed approximations must be explained and justified. Mistakes and inconsistent numbers must be corrected: they add doubts in the validity of interpretations.

Detailed remarks and comments, along the text:

- Vibrational Relaxation (page 4): W(CO)₆ vibrational relaxation in n-hexane is especially studied in ref. 56, and two characteristic times were obtained (10-20 ps and 140 ps). Is there an explanation to the missing short component in the present results? There is a specific behaviour at short times in Figure 2 which could be compared to Figure 5 (Rabi oscillations).

- Transient Spectroscopy (page 5): the spectra displayed in Figure 3 seem to be directly related to the measured transmission spectra and not transient absorption spectra that need measurement of transmission and reflexion, it must be clarified (plot $-\Delta T/T_0$, for example); replace " $1+r^2\exp...$ " by " $1+r^4\exp...$ " and " t^2 " by " t^4 " in Eq. 1 (same error in the Supplementary Information);

- Page 6, lines 122-123: "The choice of oscillators... is critical...": the authors must clarify how many oscillators they use in calculations, and must provide the used parameters describing these oscillators (in Supp Info); in particular, what are the assumptions used for the oscillator strengths of the various transitions? The calculated transmission spectrum for the ground state could be superposed to an experimental one to strengthen the analysis.

- Page 7, lines 147-149: the increase in transmission is not obvious; the authors could quantify the coupling between the cavity mode and ω_{12} with 1% of population in the excited state to prove its reality; the comparison between transient spectra with different concentrations could be useful to analyse the data (Figure 3a and Figure 6a show different behaviours - to be explained).

- Time-Dependence (page 8): the kinetic model is not presented: it seems that the population of LP is neglected whereas this state of lowest energy should be populated from UP and from the uncoupled $v=1$, as observed in other systems (refs 22, 47, for example); why a decay of UP only to the uncoupled state? The fact that only two characteristic times can describe the kinetics is not a proof of the model. The curves displayed in Figure 4 (error bars missing) are not explained in terms of populations probed at each wavenumber.

- Page 9, line 192: the "modest change" should be quantify

- Page 10, line 235: are the values of 3 and 9 ps measured, estimated? How?

- The angle dependence of the spectra is described in the Discussion, but could constitute another part of the Results.

- Page 12, line 267: 3 ps in the text and 5 ps in Figure 6

- Page 22, lines 539-540: replace w by ω

- Page 24, Figure 5a: $\Omega=23$ cm⁻¹, but $=24$ cm⁻¹ page 4, line 77; $\Omega=36$ cm⁻¹, but $=32$ cm⁻¹ page 6, line 133

- Supplementary Information: ϵ_1 instead of ϵ_2 in the first term of eq. 4

Consequently, some of the key conclusions, such as "they present "the first measurements of the population lifetime of vibration-cavity polariton" (line 299), merit to be confirmed by a better argued analysis of the data. All the features observed in transient spectra are not consistent with the present simulations and the weakness of the model (as written in the paper without clear values) does not help to believe in it. Conclusions on the observed process cannot be very general with this analysis and their specific sample.

Reviewer #3 (Remarks to the Author):

The paper reports pump-probe infrared spectroscopy of antisymmetric (C-O) stretching of $W(CO)_6$ coupled to an optical cavity. The authors had demonstrated and calculated the angle dependent transmission at different probe frequencies by varying the angle of incidence of the pump pulse to the normal of the cavity wall. From these results and using Eqn 2-4, they assign a fast decaying component to upper polariton of 14ps. They use a fairly simple fitting method of simultaneously solving two first order reactions. The topic is timely and of broad interest however, the paper has serious flaws and cannot be accepted for publication in the present form.

1. The theoretical model is not clearly defined. How do they model the reservoir? How many modes are included in the model?

2. In fig 3b there is polariton splitting of $v=1$ but not of $v=2$. This is not justified

3. Does the polariton splitting depend on the pump intensity or is it a vacuum splitting? The authors claim that cavity quantum electrodynamics is crucial here. However, the way the experiments are done, it seems questionable whether the quantum nature of the field is important. The question is if the experiment could be also described in terms of classical fields where the splitting is caused by the strong pump.

4. On p5 the authors state: "Exciting the fundamental mode depopulates the vibrational ground state which reduces the Rabi splitting". This statement is misleading since the Rabi splitting, to first order, is not a function of the vibrational excitation. The observed splitting would become smaller for higher vibrational quantum, but this does not seem to be included in their model.

5. On p10 the authors claim: "This notable observation is the first evidence for Rabi oscillations in a coupled vibration-cavity polariton system, but we note that interference between two fields at the sample (polarization beats) can also create such oscillations." The central claim of the paper, that they observed Rabi oscillation, is even questioned by the authors within the same sentence. Moreover, there might be other possible reasons for the observed oscillations, e.g., the neglected cavity coupling between $v=1$ and $v=2$ or the neglect influence of the degenerate vibrational modes and coupled vibration-cavity polaritons.

6. The statement made in page 13 "We observe no direct evidence of the transitions recently calculated by Saurabh and Mukamel" is puzzling. That reference calculated 2DIR not pump probe so it looks at different states. Are they claiming that that reference is wrong? If so, they need to spell out clearly what are their reservations. Ref 58 is the first prediction of time domain measurement of vibrational polaritons in a cavity. It should be cited as such.

Reviewers' comments:

Reviewer #1 (Remarks to the Author):

The authors present the first pump-probe study of vibro-polaritons which display impressive Rabi-oscillations at short time scales and overall modification of the dynamics. The study is interesting, sound, well written and definitely merits publication in Nature Comm., Nature Physics or PRL.

However there are some points that need clarification. Below are some specific points and comments:

1) The most important point is the way the transient spectra are recorded which is not indicated in the text. It appears that the absorbance are only acquired from the delta Ts. As was shown in ref 47, the transient Absorption spectra of sample inside cavities (whether strongly coupled or not) can only be measured accurately by recording both the reflection and the transmission. Relying on a fit using eq. 1 is not enough.

We have clarified on p. 5 that the spectra and kinetics presented in the main text are in terms of $-\log(T/T_0)$, which, the reviewer correctly points out, is not equivalent to ΔA . We have performed a series of new measurements of transient reflection spectra in tandem with transient transmission spectra in order to provide spectra and kinetic traces in units of ΔT , ΔR , and ΔA . These additional experimental and modeling results have been added as Supplementary Note 2 and are discussed in the second paragraph in the “Transient Spectroscopy of Cavity-Coupled $W(CO)_6$ ” section. We find that the decay kinetics extracted from $-\log_{10}(T/T_0)$ and ΔA are the same within the fitting uncertainties. We speculate that the effect observed in ref 37 could be system dependent and conclude that measuring ΔA is an important check, especially in this early stage of the field. We have also modeled the system using a transfer-matrix approach, described in Supplementary Note 2, rather than the simple classical method of including absorbing oscillators in the Fabry Perot cavity and find excellent agreement between calculated and observed ΔA spectra.

2) In small cavities, the large surface to volume ratios can perturb dynamics. The authors should do a control in a detuned cavity. I don't expect any change but since this is the first study of this kind, I think it is important that all the controls are done.

We have previously attempted to study cavities completely detuned from resonance with the molecular vibration and have observed no transient response from the system. To address the effects of surface-to-volume ratio and interactions with the mirror surface, we have used the same spacer thickness (25 μm) for all experiments, cavity or windowed cell. We tend to think of the surface-to-volume ratio as small in this cavity, however, to fully address this question, we have measured the transient response in a “half-cavity” comprising one cavity mirror and one CaF_2 window and have attached the results of the “half-cavity” below. We do not observe any deviation from literature values of the uncoupled dynamics in the half-cavity or relatively short path-length windowed cells (see Fig. 1 below) which verifies that the cavity surfaces are not impacting molecular kinetics.

Figure 1 | Transient absorption of $W(CO)_6$ measured at 1968 cm^{-1} in a traditional cell comprising CaF_2 windows (red, transmission) and in a half-cavity comprising one CaF_2 window and one cavity mirror (blue, transmission). Solid lines are exponential fits to the measured decays

3) The possibility of weak coupling effects which could influence the results cannot be excluded. It would be really interesting if the dynamics were studied at low enough concentration to avoid strong coupling but in resonance with the cavity.

We agree that the low-concentration experiments are interesting, but beyond the scope of this first work. We plan to systematically study the dynamics with varying concentration in the near future.

4) The authors must provide information on the composition of the mirrors and cavity response (Q factor).

We have included this information in the revised Methods section. New text reads, “The mirrors are 25 mm x 2 mm CaF_2 flats coated by Universal Thin Film Labs. We specified the proprietary coating to give $92 \pm 2\%$ reflectivity between 4.7 and 5.1 μm at normal incidence, transmission in the visible (coating is yellow to the eye), and compatibility with water and organic solvents.” We have also included the FTIR transmission spectrum of an empty cavity in the Supplemental Information.

5) The notion that there is a large reservoir of uncoupled molecules is not shared by everyone in the community. There are always some uncoupled vibrations for simple reasons such as field strength and orientation versus the field. Hence I find the conclusion of the discussion on p12 very reasonable.

We agree with this assessment and appreciate that the question of reservoir modes is unresolved. Our evidence for an angle-invariant transition at 1968 cm^{-1} (shown in the “Angle-Dependent Polariton Tuning” section) points to the presence of reservoir modes. Furthermore, the dominant feature in our measured transient spectra is analytically reproduced only with the inclusion of an oscillator at 1968 cm^{-1} and assuming that there is little ($<3\text{ cm}^{-1}$) splitting in $v=2$. The only reasonable transition at this frequency is from reservoir $v=1$ to $v=2$.

6) minor points:

- not everyone is familiar with mODs (p9 last line), so perhaps it is better just to stick to the word absorbance, e.g. 10^{-2} absorbance units.

We have now corrected the figures to make clear that the data are differential transmission (discussed throughout the Results section) and changed the specific text identified to read “would result in $-\log_{10}(T/T_0) = 0.01$.”

- The direct excitation to $v=2$ is not formally allowed by selection rules (p4), perhaps a sentence explaining why it is possible to populate this level in your system would be welcome, e.g. sequential excitation?...

The excitation pulse can interact multiple times with the sample. We have expanded on the explanation on page 4 and included a reference to a relevant article by inserting “the $W(CO)_6$ interacts with the excitation pulse more than once.³⁷”

Reviewer #2 (Remarks to the Author):

This paper reports new experimental results in the promising field of vibrational polaritons. The authors performed infrared pump-probe experiments on samples exhibiting strong coupling between vibration and cavity modes. Key results are measurements of transient spectra at the femtosecond time scale and spectrally resolved kinetics, allowing an analysis of polariton dynamics.

These original results highlight the effects of polaritons on molecular dynamics. This kind of novel experiments opens the way to novel chemistry with a possible control of chemical reactivity by coupling vibrational modes with cavity modes. In that sense, this work presents a great interest. The context is well introduced and the paper is clearly written. The T1u mode of W(CO)₆ has a very large transition dipole moment and seems a good choice to get polaritons in the strong coupling regime. Vibrational dynamics in different solvents out of a cavity was extensively studied, bringing a solid background to the present work. The quality of data as shown in the figures is very good.

However, the analysis is not convincing enough to be published in the present form. It impacts the reliability of the conclusions. First of all, several important quantitative data (results and parameters in the fits) are missing - to be added either in the manuscript or in the Supplementary Information. I detail them below. The kinetic model is especially not convincing. The assumed approximations must be explained and justified. Mistakes and inconsistent numbers must be corrected: they add doubts in the validity of interpretations.

Detailed remarks and comments, along the text:

- Vibrational Relaxation (page 4): W(CO)₆ vibrational relaxation in n-hexane is especially studied in ref. 56, and two characteristic times were obtained (10-20 ps and 140 ps). Is there an explanation to the missing short component in the present results? There is a specific behaviour at short times in Figure 2 which could be compared to Figure 5 (Rabi oscillations).

Recent work by the Hamaguchi group (ref. 41) ascribes a fast component, presumably equivalent to that reported by Heilweil and coworkers, to an anisotropic decay, i.e. a decay that arises from rotational reorganization or diffusion of the W(CO)₆. In our case, we measure the uncoupled system under "magic angle" conditions which are insensitive to these anisotropic decays. To clarify this point, we have specified that the uncoupled measurements were performed under magic angle conditions, described the implications, and included the appropriate reference in the revised Methods section.

- Transient Spectroscopy (page 5): the spectra displayed in Figure 3 seem to be directly related to the measured transmission spectra and not transient absorption spectra that need measurement of transmission and reflexion, it must be clarified (plot $-\Delta T/T_0$, for example); replace " $1+r^2\exp...$ " by " $1+r^4\exp...$ " and " t^2 " by " t^4 " in Eq. 1 (same error in the Supplementary Information);

We have clarified that we are measuring $-\log_{10}(T/T_0)$ and modified the text and axis labels appropriately. Further, we have included transient reflection and absorption spectra and kinetics measured in absorption in the supplementary information to more completely describe the behavior of the system. The results of these additional measurements are all consistent with our original interpretations. We have corrected Eq. 1 to make clear that we are considering T and R, the transmitted and reflected intensities, rather than t and r, the field amplitudes.

- Page 6, lines 122-123: "The choice of oscillators... is critical...": the authors must clarify how many oscillators they use in calculations, and must provide the used parameters describing these oscillators (in Supp Info); in particular, what are the assumptions used for the oscillator strengths of the various transitions? The calculated transmission spectrum for the ground state could be superposed to an experimental one to strengthen the analysis.

We have added text to the results section (p. 6) more fully describing the calculated spectra to make clear which oscillators are required to yield agreement with the experimental results. Specifically, we write, “As detailed below, we find that Eq. 1 adequately describes the data when we assign population to a quickly decaying upper polariton and comparatively long-lived reservoir $v = 1$ and lower polariton states.”

The next few paragraphs describe in detail the rationale for the oscillators. Furthermore, we have now included Supplementary Table 1-3 listing modeling and fit parameters, both from $-\log_{10}(T/T_0)$ and ΔA data, as well as from the angle dependent data and analysis. These tables include values used for oscillator strength and width, and, as suggested by the reviewer, we have added a plot of the experimental and calculated ground-state transmission as Supplementary Fig. 1 showing strong agreement and bolstering the validity of the model we have used.

- Page 7, lines 147-149: the increase in transmission is not obvious; the authors could quantify the coupling between the cavity mode and ω_{12} with 1% of population in the excited state to prove its reality; the comparison between transient spectra with different concentrations could be useful to analyse the data (Figure 3a and Figure 6a show different behaviours - to be explained).

We have highlighted the frequency region of interest in Figures 3a and 3d with a blue shade showing the increased transmission and have added text to this paragraph quantifying the coupling between the excited-state transitions and the cavity mode. The coupling results in a splitting of the $v = 2$ state of $2-3 \text{ cm}^{-1}$. Although this is similar to the cavity and molecular linewidths, there is clearly a measurable transient response as predicted by calculation and observed in experiment.

Regarding the behaviors shown in Figs. 3a and 6a, qualitatively, the behaviors are consistent (e.g., positive bump at the reservoir $v = 1$ state and an evolution from positive to negative response on the low energy side of the LP peak). The small differences in amplitude are due to the specific cavity tunings and splittings observed for these two experiments. We have added steady-state transmission spectra as insets to Fig. 6a which clarify this point. We are currently working toward a detailed study of the concentration and angle dependence of the phenomena reported in this seminal work.

- Time-Dependence (page 8): the kinetic model is not presented: it seems that the population of LP is neglected whereas this state of lowest energy should be populated from UP and from the uncoupled $v=1$, as observed in other systems (refs 22, 47, for example); why a decay of UP only to the uncoupled state? The fact that only two characteristic times can describe the kinetics is not a proof of the model. The curves displayed in Figure 4 (error bars missing) are not explained in terms of populations probed at each wavenumber.

We agree with reviewer and appreciate his/her pointing out that, in the cavity, we do not know the specific relaxation mechanism and may not simply assume consecutive kinetics, unlike the case outside the cavity. We have substantially modified our analysis of the transients and our associated discussion of the kinetic model to address this point. The reviewer’s comments led us to realize that a non-sequential (i.e. biexponential) decay model also describes the dynamics and yields the same decay rates. This model is more general and represents the simplest model that accounts for our observations (this is now discussed on p. 9). We also agree that the LP state is likely populated and have performed additional calculations, which are mentioned in the text (p. 9-10) and described in detail in Supplementary Note 4 “Spectral Signature of LP Population.” The modified text now emphasizes that the slow decay we observe contains information on both LP- and reservoir-excited molecules. Unfortunately, these two states are spectrally indistinguishable in this system. We have included a representative error bar in Figure 4 and have added text to explain that each probe wavenumber reports on the evolution of multiple species because the cavity-coupled spectrum of each species spans the entire frequency range of the measurement. All of these issues are discussed on p. 8-10.

- Page 9, line 192: the "modest change" should be quantify

We have quantified the change to the Rabi splitting by stating "modest ($<1\text{ cm}^{-1}$, see Fig. 3c)" on p. 10.

- Page 10, line 235: are the values of 3 and 9 ps measured, estimated? How?

We have added to the text an explanation that we estimate the dephasing times from the homogeneous linewidths of the cavity and vibrational modes.

- The angle dependence of the spectra is described in the Discussion, but could constitute another part of the Results.

We thank the reviewer for the suggestion and have moved the angle dependence into the Results section (p. 10), where it now immediately follows the presentation of the kinetic traces.

- Page 12, line 267: 3 ps in the text and 5 ps in Figure 6

We have corrected the value in the text to reflect the correct value, 5 ps.

- Page 22, lines 539-540: replace ω by Ω

We have fixed this error and thank the reviewer for pointing it out.

- Page 24, Figure 5a: $\Omega = 23\text{ cm}^{-1}$, but $= 24\text{ cm}^{-1}$ page 4, line 77; $\Omega = 36\text{ cm}^{-1}$, but $= 32\text{ cm}^{-1}$ page 6, line 133

We have corrected the erroneous value of 36 cm^{-1} in the text to the accurate value, 34 cm^{-1} . Small variations in sample concentration and angle lead to different Rabi splittings. We have added an explanation of this small sample-to-sample variation to the Methods section.

- Supplementary Information: ϵ_1 instead of ϵ_2 in the first term of eq. 4

We have corrected this typographic error and thank the reviewer for bringing it to our attention. Calculations were performed with the proper equations. We have confirmed the error was in translation to the text of the SI.

Consequently, some of the key conclusions, such as "they present "the first measurements of the population lifetime of vibration-cavity polariton" (line 299), merit to be confirmed by a better argued analysis of the data. All the features observed in transient spectra are not consistent with the present simulations and the weakness of the model (as written in the paper without clear values) does not help to believe in it. Conclusions on the observed process cannot be very general with this analysis and their specific sample.

We appreciate the reviewer's concerns and have substantially modified the Results section to clarify and expand on the model we use to describe the results. Because of the coincidental resonance between ω_{01} and $\omega_{LP\ 10\ 2}$, we cannot determine the LP lifetime in this system, but we think that systems where the anharmonicity does not match the experimentally accessible Rabi splitting may be appropriate to address this open question. We have obtained preliminary data on thiocyanate (NCS^-) and dicyanamide ($\text{N}(\text{CN})_2^-$) in aqueous and methanol solution which show qualitatively similar behavior to $\text{W}(\text{CO})_6$. We hope to soon show that the conclusions in the present work are applicable to these and other systems.

Reviewer #3 (Remarks to the Author):

The paper reports pump-probe infrared spectroscopy of antisymmetric (C-O) stretching of W(CO)₆ coupled to an optical cavity. The authors had demonstrated and calculated the angle dependent transmission at different probe frequencies by varying the angle of incidence of the pump pulse to the normal of the cavity wall. From these results and using Eqn 2-4, the assign fast decaying component to upper polariton of 14ps. They use fairly simple fitting method of simultaneously solving two first order reactions. The topic is timely and of broad interest however, the paper has serious flaws and cannot be accepted for publication in the present form.

1. The theoretical model is not clearly defined. How do they model the reservoir? How many modes are included in the model?

We have substantially modified and clarified the description of the models used to calculate both the transient spectra and fit the dynamics. Specifically, we have stated which modes are used to calculate the spectra (p. 6-7), expanded the model description in the Supplementary Information (Supplementary Notes 1 and 3 and Tables 1-3), and included references to previous studies (ref. 19,46,47. p. 6-7) in which reservoir modes are observed with both vibrational and electronic transitions.).

2. In fig 3b there is polariton splitting of $v=1$ but not of $v=2$. This is not justified

The vacuum Rabi splitting depends on the square root of absorber concentration in the lower level of the transition. Because we excite only about 1% of the molecules, $N^{1/2}$ is 10 times smaller for the excited-state transitions than the ground-state transition. Consequently, the excited-state splitting is ~10 times smaller, (~2-3 cm^{-1}). We have added text to the Results section (p. 7) to explain this and have modified Fig. 3b to make this clearer.

Quantum mechanical treatments of strongly coupled higher excited states indicate that they exhibit strong splitting (see, for one example, energy level diagram in ref. 32), which arise due to multiphoton interactions. If these states were present in our system, one would observe a strong transition between the lower $v = 1$ polariton and the highest $v = 2$ polariton, which we calculate to reside at 1998 cm^{-1} . We observe no evidence of this transition and suspect that this “ladder climbing” could be more readily observed in coherent spectroscopies such as the two-dimensional infrared spectra calculated in ref. 32.

3. Does the polariton splitting depend on the pump intensity or is it a vacuum splitting? The authors claim that cavity quantum electrodynamics is crucial here. However, the way the experiments are done, it seems questionable whether the quantum nature of the field is important. The question is if the experiment could be also described in terms of classical fields where the splitting is caused by the strong pump.

We do not claim that cavity quantum electrodynamics is crucial. The splitting we observe is well established as a vacuum splitting for one oscillator multiplied by a factor of $N^{1/2}$ to obtain an effective splitting (on p. 2, see especially ref 55, section 5 and refs 26 and 27 for the first demonstrations with vibrational modes). We observe the same splitting whether measured in an FTIR spectrometer with very weak IR intensity or with intense ultrafast infrared pulses, verifying that the splitting we observe does not depend on pump intensity.

4. On p5 the authors state: "Exciting the fundamental mode depopulates the vibrational ground state which reduces the Rabi splitting". This statement is misleading since the Rabi splitting, to first order, is not a function of the vibrational excitation. The observed splitting would become smaller for higher vibrational quantum, but this does not seem to be included in their model.

We disagree with the comment that the splitting does not depend on the degree of vibrational excitation. It is well established for electronic transitions (refs 15, 55 for reviews) and vibrational transitions (refs 26, 27, 28, 29) that the effective Rabi splitting is directly proportional to the square root of the absorber concentration, i.e., $\Omega = g(N^{1/2})$. As such, generating an excited population reduces the concentration of ground-state molecules, and thus, the observed splitting. Indeed, studies on electronic transitions have shown that depopulating the ground state with an excitation pulse reduces the splitting (refs 37, 38, 44). We find the terminology adopted by the community confusing, since the ensemble effective splitting is easily confused for what others call Rabi splittings (as noted on p. 12 regarding uncoupled oscillation, ref. 59) that depend on pump power, etc. We have added the “vacuum” or “effective” qualifier in many instances in the text when discussing Rabi oscillations and splitting to clarify the phenomenon we are examining in this work.

5. On p10 the authors claim: "This notable observation is the first evidence for Rabi oscillations in a coupled vibration-cavity polariton system, but we note that interference between two fields at the sample (polarization beats) can also create such oscillations." The central claim of the paper, that they observed Rabi oscillation, is even questioned by the authors within the same sentence. Moreover, there might be other possible reasons for the observed oscillations, e.g., the neglected cavity coupling between $v=1$ and $v=2$ or the neglect influence of the degenerate vibrational modes and coupled vibration-cavity polaritons.

While we don't agree that the observation of Rabi oscillations is the single central claim of our work, we have strengthened our claims with additional evidence that the observed features are Rabi oscillations. Specifically, these oscillatory features are observed in both parallel- and cross-polarized conditions (i.e., the probe beam polarization either parallel or perpendicular to the pump). The text on p. 12 now includes this information and more confidently makes the case that we observe vacuum Rabi oscillations. Furthermore, the evidence we present goes beyond that used in ref. 45 to identify this effect, since we show that the oscillation frequency tracks the splitting over the concentration range with which we achieve strong coupling.

6. The statement made in page 13 "We observe no direct evidence of the transitions recently calculated by Saurabh and Mukamel" is puzzling. That reference calculated 2DIR not pump probe so it looks at different states. Are they claiming that that reference is wrong? if so, they need to spell out clearly what are their reservations Ref 58 is the first prediction of time domain measurement of vibrational polaritons in a cavity. It should be cited as such.

We appreciate that the article by Saurabh and Mukamel is breaking new ground in the field. We have moved its mention to the introduction and pointed out its position as the first theoretical work describing 2DIR of vibration-cavity polaritons. We have removed the specific line referenced by the reviewer, realizing that the double quantum coherence method is different enough from our pump-probe spectroscopy that direct comparisons might not be appropriate. Our original comment was meant to point out that we do not observe transitions to the states one would expect from large $v = 2$ splitting, as described in the response to this reviewer's second point.

Reviewers' comments:

Reviewer #1 (Remarks to the Author):

The authors have answered the reviewers comments in a very satisfactory manner and I recommend acceptance as is.

Since the field is moving very fast, the authors might consider adding for the benefit of the readers a reference to a recent article on the significant modification of chemical reactivity under vibrational strong coupling: Thomas et al *Angewandte Chemie Int. Ed.* 10.1002/anie.201605504

Reviewer #2 (Remarks to the Author):

The revised version of the manuscript shows large improvements. Results are better explained, some assertions are clarified. Additional data were collected. However, the analysis of the time-dependent data needs further improvements to be published. The model is still to be clarified and its consistency is not obvious.

1) Transient spectra: I understand that the authors prefer to discuss transient transmission than transient absorption with a model based on Equation 1. I guess that they fix the amount of excited molecules (1%, i.e. a surprising low value) from the fit of the higher part of the spectrum (around 2000 cm^{-1}), where no additional transition occurs, reflecting only the decrease of the effective splitting (with 1%, this reduction is much less than 1 cm^{-1} , far below the bandwidths). In this range, the experimental data do not show large effects, and fits are not conclusive. In the other spectral range (around 1965 cm^{-1}), there are many possible transitions, so that modelling is a hard task. Anyway, the agreement between the calculated and the experimental spectra is correct (not "excellent, as claimed by the authors, page 8 line 179; for example, the large negative part around 1973 cm^{-1} is never fitted). Consequently, the authors obtain two sets of oscillators to be included in Eq. 1 to reproduce the two spectra of Figure 3a. I suspect they use the parameters written in Supplementary Note 2. It is not clearly written. The change in absorber population is 0.25% in Supplementary note 2 and note 4, and 1% in the text; moreover, the authors use "Oscillator strengths" in cm^{-2} , whereas there are no units for oscillator strengths, they play with the amplitudes of the Lorentzian lines which include oscillator strengths and populations. As the oscillator strengths of transitions involving polaritons are unknown (probably depending on their vibrational component), it is difficult to give populations (whereas it is given page 10, line 213). The best is to keep amplitudes. In these conditions, conserving a constant total of "Oscillator strengths" (as written in Fig. S2) is not obligatory.

The spectral range around 1961 cm^{-1} is particularly discussed. The model shows tendencies but fails to reproduce all the features in a semi-quantitative way. It must be noted that the experimental signal is very low and positive features are not far from negative ones. In this spectral range, there is a dark line in Fig. 1b which could be a signature of another resonance: what is it? And what are the consequences of this physical(?) effect in the time-resolved spectra?

2) Kinetics: From the spectra modelling, the positive part disappearing at $t=100\text{ps}$ around 1961 cm^{-1} is due to the UP to 2 transition and the authors conclude from this fact (only this one, it seems) that the UP state has a short lifetime. On the other hand, the authors reproduce the time-evolutions of the signals by bi-exponential decays. They conclude that the shortest characteristic time corresponds to the lifetime of the UP polariton. Such a conclusion needs a real kinetic model involving the populations of all the levels at play. The equations S6-S8 of Supplementary Note 5 cannot reproduce the kinetics: LP population is missing, population transfers between UP, $v=1$ and LP are missing...

As the authors can reproduce the transient spectra, they could also try to reproduce the decay curves of Figure 4, using Eq.1 and time-dependent amplitudes of Lorentzians. Did they try to do it with Eq. S6-S8? I would be interested by the result.

Parameters in Tables 1-3 of Supp. Info are not commented

In conclusion, the authors should be more cautious in the conclusions they obtain from their model. For example, they cannot claim "The value determined for k_{UP} represents the first direct measurement of a vibrational polariton lifetime" or "These results represent the first measurements of the population lifetime of a vibration-cavity polariton" because the value they give cannot be assigned without doubt to this lifetime. Their experimental results are new, original and of good quality, but the confuse analysis serves badly the publication in the present form.

Additionally, there are some discrepancies between the responses to the reviewer's comments and the text: is it the last version?

Details:

-page 4, line 88: include ref 42 in refs 39-41

-page 8, line 178: "Note 4" instead of "Note 3"

-Figure 3a: add the "Colored vertical lines on the frequency axis" corresponding to probe frequencies presented in Figure 4" as written in the caption and in the text.

-check the numbering of Figures and Notes

- "Discussion" contains mainly concluding remarks and can be changed to "Conclusion", while "Results" is "Results and discussion"

- page 13, line 310: the authors mention the proximity of acceptor modes to justify the change in lifetimes when detuning the cavity: could they specify which modes are possible acceptor modes?

Reviewer #3 (Remarks to the Author):

The authors have adequately addressed the issues raised in my previous report. The paper is greatly improved.

The work is novel and timely and I recommend it be accepted for publication in Nature Communications

Reviewer #1 (Remarks to the Author):

The authors have answered the reviewers comments in a very satisfactory manner and I recommend acceptance as is.

Since the field is moving very fast, the authors might consider adding for the benefit of the readers a reference to a recent article on the significant modification of chemical reactivity under vibrational strong coupling: Thomas et al *Angewandte Chemie Int. Ed.* 10.1002/anie.201605504

We agree with the reviewer and have included a reference to this exciting article which we became aware of after submission. We thank the reviewer for the recommendation to publish.

Reviewer #2 (Remarks to the Author):

The revised version of the manuscript shows large improvements. Results are better explained, some assertions are clarified. Additional data were collected. However, the analysis of the time-dependent data needs further improvements to be published. The model is still to be clarified and its consistency is not obvious.

1) Transient spectra: I understand that the authors prefer to discuss transient transmission than transient absorption with a model based on Equation 1. I guess that they fix the amount of excited molecules (1%, i.e. a surprising low value) from the fit of the higher part of the spectrum (around 2000 cm^{-1}), where no additional transition occurs, reflecting only the decrease of the effective splitting (with 1%, this reduction is much less than 1 cm^{-1} , far below the bandwidths). In this range, the experimental data do not show large effects, and fits are not conclusive. In the other spectral range (around 1965 cm^{-1}), there are many possible transitions, so that modelling is a hard task.

There is a misunderstanding here. We actually use the magnitudes of the strong features near 1968 and 1961 cm^{-1} as indicators of good agreement with the calculations. We have updated the text on page 10 to include, “especially those of the features at 1968 cm^{-1} and 1961 cm^{-1} ,” to clarify this issue. We emphasize that, as described in the text, these traces are not fits but representative spectra.

Anyway, the agreement between the calculated and the experimental spectra is correct (not “excellent, as claimed by the authors, page 8 line 179; for example, the large negative part around 1973 cm^{-1} is never fitted).

We respectfully disagree with the reviewer with regard to the feature at 1973 cm^{-1} . Our calculated spectra (Fig 3a, 3d) do predict a negative feature in this region which arises from both the reduced splitting of the ground-state transition and coupling to the reservoir excited-state transition.

Consequently, the authors obtain two sets of oscillators to be included in Eq. 1 to reproduce the two spectra of Figure 3a. I suspect they use the parameters written in Supplementary Note 2. It is not clearly written. The change in absorber population is 0.25% in Supplementary note 2 and note 4, and 1% in the text; moreover, the authors use “Oscillator strengths” in cm^{-2} , whereas there are no units for oscillator strengths, they play with the amplitudes of the Lorentzian lines which include oscillator strengths and populations.

We thank the reviewer for pointing out several inconsistencies, especially in the Supplementary Information. We inadvertently switched between describing total excitation and excitation residing in individual excited states and have now changed each description to consistently reference the total excitation (i.e. reduction in ground-state Lorentzian amplitude, p.23 (Fig. 3 caption) and p. 6). In

addition, we have corrected all instances of “oscillator strength” to instead read “Lorentzian amplitude” or “amplitude.” (p. 7, S5, S6)

As the oscillator strengths of transitions involving polaritons are unknown (probably depending on their vibrational component), it is difficult to give populations (whereas it is given page 10, line 213). The best is to keep amplitudes. In these conditions, conserving a constant total of “Oscillator strengths” (as written in Fig. S2) is not obligatory.

We agree that the oscillator strengths of polariton excited-state absorptions are, as yet, unknown, but believe the most reasonable assumption is that they are equal to the reservoir $v = 1$ to 2 oscillator strength. We have added a sentence to page 7 to specify this assumption, by stating “The total oscillator amplitude in the system is conserved and we assume that the polariton excited-state absorption transition intensity is the same as for the reservoir transition”. Also, to further address this concern we have inserted a sentence on page 10 that reads, “If coupling to the cavity modifies the excited-state absorption intensity of the polaritons compared to the reservoir, our estimate of the polariton population will be subsequently modified.”

The spectral range around 1961 cm^{-1} is particularly discussed. The model shows tendencies but fails to reproduce all the features in a semi-quantitative way. It must be noted that the experimental signal is very low and positive features are not far from negative ones.

We reproducibly see the transition from positive to negative across months of measurements. Transient decays at low probe energies ($<1960 \text{ cm}^{-1}$) clearly illustrate this transition.

In this spectral range, there is a dark line in Fig. 1b which could be a signature of another resonance: what is it? And what are the consequences of this physical(?) effect in the time-resolved spectra?

The dark line arises from weak coupling to the very small absorption feature of naturally occurring singly ^{13}C substituted $\text{W}(\text{CO})_6$. We speculate this feature could contribute to the anomalous behavior in Fig 6b and plan experiments to address that question in the future.

2) Kinetics: From the spectra modelling, the positive part disappearing at $t=100\text{ps}$ around 1961 cm^{-1} is due to the UP to 2 transition and the authors conclude from this fact (only this one, it seems) that the UP state has a short lifetime. On the other hand, the authors reproduce the time-evolutions of the signals by bi-exponential decays. They conclude that the shortest characteristic time corresponds to the lifetime of the UP polariton. Such a conclusion needs a real kinetic model involving the populations of all the levels at play. The equations S6-S8 of Supplementary Note 5 cannot reproduce the kinetics: LP population is missing, population transfers between UP, $v=1$ and LP are missing...

We generally agree with this assessment, but two key factors motivate our chosen kinetic analysis.

First, as we described in the SI, there is no distinct spectral signature from LP population because of the similarity between the anharmonic shift and the effective splitting.

Second, spectral contributions from the UP and reservoir are not separable. We are unable to distinguish between, for instance, the UP decaying to the solvent bath or decaying into the $v = 1$ reservoir (see calculated kinetics in response to issue below – what looks like a rise in the reservoir response can also be explained by a decay in the UP response).

To address the reviewer’s comment, we have added a paragraph to the discussion. It reads, “The simple biexponential kinetic model we employ is somewhat surprising in its effectiveness. In other polariton systems, incoherent energy transfer between the polariton modes and reservoir modes has been

observed.¹⁹ In this case, the congested spectra preclude us from distinguishing between these pathways. Each excited-state absorption transition couples to the cavity, leading to overlapping spectral responses. There is no spectral region where the transient response reports only, for instance, on the reservoir population, so we cannot distinguish between the model presented above (UP and reservoir both decay independently) or a decay path in which the UP decays to the reservoir. Other molecular systems where the anharmonicity is very different from experimentally accessible splittings might give clear evidence of more complicated dynamics.”

As the authors can reproduce the transient spectra, they could also try to reproduce the decay curves of Figure 4, using Eq.1 and time-dependent amplitudes of Lorentzians. Did they try to do it with Eq. S6-S8? I would be interested by the result.

We have done this, but think that presenting them in the manuscript or SI is not informative and detracts from our main result. Results from this kind of analysis are shown in Fig. R1. The red circles are the experimental data from Fig 4 and the blue line is the calculated response. We calculate the response by including time-dependent amplitudes (Eqs. S6-S8) in Eq. 1. We use the parameters we report in Supplementary Note 5 (at the bottom of p. S6) for the initial amplitudes, total amplitude, and widths. Differences in laser performance between when the data in Fig 3 (from which the initial amplitudes are estimated) and those in Fig 4 were measured lead to differences in the signal magnitude, so we normalize the traces. The calculation shows that Eqs. S6-S8 can reproduce the functional form of the observed kinetics. We offer the caveat that we did not fully explore using Eq. 1 to describe the kinetics.

Figure R1 | Comparison between calculated and measured kinetic response. Transient response of 20 mM $\text{W}(\text{CO})_6$ in hexane coupled to a cavity measured at 1970 cm^{-1} . Red circles are data (see also Fig. 4), blue line is the calculated trace described in the text above.

We still agree with the reviewer that other relaxation pathways might or should exist, but this system appears to be inappropriate for discerning those pathways. We are currently designing experiments on other molecules where, for instance, the anharmonicity is much larger. In that case, it should be easier to observe, identify, and distinguish responses from the LP and UP.

Parameters in Tables 1-3 of Supp. Info are not commentated

We have expanded on the captions for Tables S1 and S3. For all three tables, relevant commentary on the parameters is present in the text associated with the tables (either in manuscript or SI). The captions are consistent with other table captions in Nature Communications articles.

In conclusion, the authors should be more cautious in the conclusions they obtain from their model. For example, they cannot claim “The value determined for kUP represents the first direct measurement of a vibrational polariton lifetime” or “These results represent the first measurements of the population lifetime of a vibration-cavity polariton” because the value they give cannot be assigned without doubt to this lifetime. Their experimental results are new, original and of good quality, but the confuse analysis serves badly the publication in the present form.

We respectfully disagree with the reviewer’s assessment of the UP assignment. We have shown that the transient spectra are consistent with the UP to 2 absorption frequency and, crucially, that this frequency and decay rate change with incidence angle. Such behavior is only consistent with polariton modes.

We have modified the first statement (on p. 9) pointed out by the reviewer to indicate that the angle-dependence of the lifetime substantiates this assignment.

We are very grateful to the reviewer for the many incisive comments and think that our revisions have made our analysis, especially with regard to the limitations in the kinetic analysis of this specific system, more clear.

Additionally, there are some discrepancies between the responses to the reviewer’s comments and the text: is it the last version?

Details:

-page 4, line 88: include ref 42 in refs 39-41

We have corrected this oversight and thank the reviewer for bringing it to our attention.

-page 8, line 178: “Note 4” instead of “Note 3”

We have corrected this error and thank the reviewer for bringing it to our attention.

-Figure 3a: add the “Colored vertical lines on the frequency axis” corresponding” to probe frequencies presented in Figure 4” as written in the caption and in the text.

We have replaced the vertical lines that we inadvertently removed from a previous version of the figure.

-check the numbering of Figures and Notes

We have corrected the numbering in the SI, rearranged the Supplementary Tables to conform to Nature Communications guidelines, and appropriately adjusted in-manuscript references. We thank the reviewer for the careful attention paid.

- “Discussion” contains mainly concluding remarks and can be changed to “Conclusion”, while “Results” is “Results and discussion”

The paragraph we added in response to a comment above helps to address this issue, but we believe the section labels we use are consistent with Nature Communications guidelines.

- page 13, line 310: the authors mention the proximity of acceptor modes to justify the change in lifetimes when detuning the cavity: could they specify which modes are possible acceptor modes?

We have added a sentence reading, “In metal carbonyls, these acceptor modes can include slightly higher-lying Raman active modes, low-frequency bending modes of the metal carbonyl, localized solvent vibrations, and collective solvent motion.^{40,50-54}”, to expand on this question.

Reviewer #3 (Remarks to the Author):

The authors have adequately addressed the issues raised in my previous report. The paper is greatly improved.

The work is novel and timely and I recommend it be accepted for publication in Nature Communications

We thank the reviewer for the recommendation to publish and for the previous insightful comments.

REVIEWERS' COMMENTS:

Reviewer #2 (Remarks to the Author):

The revised version of the manuscript fully answers the latest recommendations. The assumptions made in the analysis of the experimental results are now clearly explained and discussed.

Because of the novelty and the quality of the work, I recommend the paper for publication in Nature Communications.

Reviewer #2 (Remarks to the Author):

The revised version of the manuscript fully answers the latest recommendations. The assumptions made in the analysis of the experimental results are now clearly explained and discussed.

Because of the novelty and the quality of the work, I recommend the paper for publication in Nature Communications.

We thank the second reviewer for the recommendation to publish and for the incisive and helpful comments provided in earlier rounds of review.